# Microfiltration Membranes for the Removal of Bisphenol A from Aqueous Solution: Adsorption Behavior and Mechanism

**Jiaoxia Sun [1,\*], Xueting Jiang [1], Yao Zhou [1], Jianxin Fan [1] and Guoming Zeng [2]**

[1] School of River and Ocean Engineering, Chongqing Jiaotong University, Chongqing 400074, China; jiangxueting@mails.cqjtu.edu.cn (X.J.); camellia5249@163.com (Y.Z.); jxfanw@gmail.com (J.F.)

[2] School of Architecture and Engineering, Chongqing University of Science and Technology, Chongqing 401331, China; 2017015@cqust.edu.cn

\* Correspondence: sjx@cqu.edu.cn

**Abstract:** This study mainly investigated the adsorption behavior and mechanism of microfiltration membranes (MFMs) with different physiochemical properties (polyamide (PA), polyvinylidene fluoride (PVDF), nitrocellulose (NC), and polytetrafluoroethylene (PTFE)) for bisphenol A (BPA). According to the adsorption isotherm and kinetic, the maximum adsorption capacity of these MFMs was PA (161.29 mg/g) > PVDF (80.00 mg/g) > NC (18.02 mg/g) > PTFE (1.56 mg/g), and the adsorption rate was PVDF ($K_1$ = 2.373 h$^{-1}$) > PA ($K_1$ = 1.739 h$^{-1}$) > NC ($K_1$ = 1.086 h$^{-1}$). The site energy distribution analysis showed that PA MFMs had the greatest adsorption sites, followed by PVDF and NC MFMs. The study of the adsorption mechanism suggested that the hydrophilic microdomain and hydrophobic microdomain had a micro-separation for PA and PVDF, which resulted in a higher adsorption capacity of PA and PVDF MFMs. The hydrophilic microdomain providing hydrogen bonding sites and the hydrophobic microdomain providing hydrophobic interaction, play a synergetic role in improving the BPA adsorption. Due to the hydrogen bonding force being greater than the hydrophobic force, more hydrogen bonding sites on the hydrophobic surface resulted in a higher adsorption capacity, but the hydrophobic interaction contributed to improving the adsorption rate. Therefore, the distribution of the hydrophilic microdomain and hydrophobic microdomain on MFMs can influence the adsorption capacity and the adsorption rate for BPA or its analogues. These consequences provide a novel insight for better understanding the adsorption behavior and mechanism on MFMs.

**Keywords:** microfiltration membranes; bisphenol A; adsorption mechanism; hydrogen bonding; hydrophobic interaction

## 1. Introduction

Water pollution has invariably been one of the focuses of the prevention of environmental pollution. In recent decades, endocrine disruptors (EDCs), the emerging pollutant detected in drinking water, which seriously affects human health and the environment, have drawn significant social and scientific concerns [1–4]. Several published studies have reported that EDCs are widely detected in effluents released from sewage treatment plants, surface water, and drinking water [5–10]. Bisphenol A, which is one of the important compounds in the production of polycarbonate plastics and resins, is a typical environmental endocrine disruptor of high concern due to its role in the use of many industrial compounds worldwide [11]. Despite its low toxicity, long-term exposure to BPA may pose health risks to the reproductive and endocrine systems of animals, and it is evident that high concentrations of BPA may pose greater health risks to developmental and neurodevelopmental toxicity in animals [12–14]. Therefore, efficient removal of BPA from water is significant for the drinking water safety.

Several methods have been reported for the removal of pollutants, including electro-oxidation, photo-oxidation, biodegradable, adsorption, and membrane technology [15–20].

In wastewater treatment, membrane technology promises a bright future with many advantages, such as relatively high efficiency, selectivity, low cost, minor footprint, and less secondary pollution [21]. Membrane treatment technology, according to the pore size of the membrane, can be generally divided into Microfiltration (MF), Ultrafiltration (UF), Nanofiltration (NF), and Reverse Osmosis (RO) [20]. Much progress has been made recently in the adsorption and enrichment treatment of organic pollutants, heavy metals, and other pollutants in water by using the membrane and the membrane modified by physical or chemical methods [22–25]. UF membranes mainly use physical screening and micro-osmosis to remove impurities in water through the pressure difference on both sides of the membrane, so as to achieve the separation of large and small substances [26]. The molecular weight of NF membranes is usually in the range of 150~2000 Da, which is between the RO membrane and the UF membrane [27]. Although NF and RO membranes represent valid removal of various nanoscale molecules and ionic solutes based on size exclusion and electrostatic interactions, the process requires high trans-membrane pressure and is liable to fouling, which affects their practical uses [28,29]. MF membranes are microporous membranes, which have the characteristics of high voidage, high throughput filtration, and no secondary pollution [30]. MF membranes can be modified by various molecules to capture target solutes [31,32]. Sieving retention by these membranes is scarcely possible because of their large pore size, but adsorption plays a key role in the removal of target contaminants in water by MF membranes [33]. The research found that estrone could be readily removed from water by nylon MF membrane due to the chemisorption that occurred between estrone and the nylon membrane [34]. Based on these studies, we hypothesized that microporous membranes with different chemical compositions may exhibit different adsorption behavior for BPA. It would provide useful information to find or design the proper MF membrane with good adsorption capacity by a deep understanding of the adsorption mechanism of MFMs for BPA.

In this work, four specific MFMs with different chemical compositions (PA, PVDF, NC, and PTFE) were selected to analyze their adsorption behavior and adsorption mechanism. They are common membrane materials and are widely used in the field of separation [35–38]. We investigated the adsorption behaviors of the after-mentioned BPA on four MFMs with respect to adsorption kinetics, isotherms, site energy analysis, and rapid filtration adsorption. The adsorption mechanism of the BPA on MFMs were studied intensively based on the physiochemical properties. By exploring the adsorption mechanism of BPA on MFMs, this study will contribute to choosing or designing MFMs as an adsorption filter membrane using the removal of specific pollutants.

## 2. Materials and Methods

### 2.1. Materials

Commercially available PA MFMs, PVDF MFMs, NC MFMs, and PTFE MFMs (average pore size = 0.22 µm) were supplied by Tianjin Keyilong Experimental Equipment Co., Ltd. (Tianjin, China). The properties of the four MFMs are shown in Table 1. BPA (99%) was purchased from Beijing Bailingwei Technology Co., Ltd. (Beijing, China). The chemical structure of BPA is shown in Figure 1. Methanol ($CH_3OH$, AR) was acquired from Sinopharm Chemical Reagent Co., Ltd. (Shanghai, China). All the reagents were used as received. The water used in all the experiments was deionized water prepared by our lab.

**Figure 1.** The chemical structure of BPA.

**Table 1.** Properties of four MFMs.

| Membrane Material | Molecular Structure | pH Tolerance | Temperature Tolerance | Performance |
|---|---|---|---|---|
| PA | | 3–10 | 100 °C | Excellent mechanical intensity, high separation number, chemical resistance, high thermal stability [39,40] |
| PVDF | | 3–10 | 100 °C | Low tensile strength, brittle membrane, resistant to dilute acid and weak base, flammable, and produces toxic oxides of nitrogen during combustion [41] |
| NC | | 3–10 | 75 °C | Preeminent chemical resistance, mechanical stability, strong negative electrostatic, good flexibility [42,43] |
| PTFE | | 3–10 | 100 °C | High water resistance, excellent electrical insulation properties, and a wide range of high and low-temperature use [44] |

Note(s): Data in Table 1 were obtained from the official website of Tianjin Keyilong Experimental Equipment Co., Ltd. (Tianjin, China).

*2.2. Adsorption Experiments*

The adsorption experiments were conducted at room temperature (25 °C). The concentration of the BPA solution was measured by an Ultraviolet-Visible spectrophotometer (UV-3150, Shimadzu, Kyoto, Japan) at 277 nm wavelength.

2.2.1. Batch Adsorption

In the adsorption kinetic experiments, the initial concentration of the BPA solution was fixed at 20 mg/L. A piece of the membrane (0.05 g) and 50 mL of BPA solutions were placed in a conical flask (100 mL). All the flasks were placed in a constant temperature oscillation box, which vibrated at 150 rpm. The samples were collected at different times. For adsorption isotherm experiments, a piece of the membrane (0.05 g) and 50 mL of BPA solutions with different concentrations (5, 10, 20, 30, 50 mg/L) were added into a conical flask (100 mL). All the flasks were placed in a constant temperature oscillation box, which vibrated at 150 rpm for 24 h to achieve equilibrium. Control experiments (without membranes) were carried out simultaneously. The concentrations of BPA measured in control experiments were used as the initial concentrations. All experiments were conducted in duplicate.

2.2.2. Rapid Filtration Adsorption

Laboratory-scale rapid filtration adsorption experiments were conducted to investigate the efficiency of membranes for removing BPA from water during the filtration process. The schematic diagram of the experimental apparatus of rapid filtration adsorption is provided in Figure S1. A piece of the membrane (0.12 g, $1.256 \times 10^{-3}$ m$^2$) was placed in the filtering device, and 50 mL of 10 mg/L BPA solution was poured into the device for filtering. The filtering was processed under nature pressure without external pressure. Filtration was collected to detect the concentration of BPA. To compare with the BPA removing efficiency under the condition of equilibrium adsorption, the equilibrium adsorption experiments were conducted by placing the same size of membranes and 50 mL of 10 mg/L BPA solution into a conical flask at 150 rpm for 24 h to reach adsorption equilibrium. Control

experiments (without membranes) were carried out simultaneously. The concentrations of BPA measured in control experiments were used as the initial concentrations. All experiments were conducted in duplicate.

### 2.3. Characterization and Analytical Methods

#### 2.3.1. Characterization Methods

The surface morphology of MFMs was observed with a field-emitting scanning electron microscope (ZEISS, Gemini 300, Jena, Germany) after being sputtered with the gold layer. The surface chemistry of the MFMs was analyzed by energy-dispersive X-ray spectrometry (EDS) conducted with an Octane elect super (EDAX, Mahwah, NJ, USA). The MFMs before and after BPA adsorption were analyzed using an FTIR instrument (IRTRACER-100, Shimadzu) equipped with a ZnSe crystal under attenuated total reflection settings (ATR-FTIR). The infrared spectra of the samples were recorded between 600 and 4000 $cm^{-1}$ using 32 scans obtained at a resolution of 2 $cm^{-1}$.

The water contact angles of the three MFMs were identified by the Video Optical Contact angle Measurement (Hake Test Instrument, Beijing, China). Each membrane was subjected to five measurements at least, and the average value was taken. To calculate the coefficient of water absorption of MFMs, the following experiments were performed in triplicate. A piece of the washed and dried membrane (0.12 g, $1.256 \times 10^{-3}$ $m^2$) was weighed and dipped into ultra-pure water (50 mL) for 48 h at 25 °C. Then, the membrane was taken out by forceps to eliminate excess water and then weighted immediately. The coefficient of water absorption of MFMs (W, %) is shown in the following Equation (1).

$$W = \frac{m - m_0}{m_0} \times 100\% \tag{1}$$

where $m_0$ (g) and m (g) denote the mass of the membrane before and after adsorption, respectively.

#### 2.3.2. Adsorption Kinetics

The adsorption kinetics can usually be characterized by mathematical models, including the pseudo-first-order model and pseudo-second-order model [45,46].

The pseudo-first-order model is provided in the following Equation (2):

$$q_t = q_e \left(1 - e^{-K_1 t}\right) \tag{2}$$

The pseudo-second-order model adsorption model can be represented as follows:

$$q_t = \frac{K_2 q_e^2 t}{1 + K_2 q_e t} \tag{3}$$

where $q_t$ (mg/g) and $q_e$ (mg/g) are the amount of adsorbent adsorbed at time t (h) and the equilibrium adsorption capacity of adsorbent, respectively; $K_1$ (1/h) is the adsorption rate constant of the pseudo-first-order; $K_2$ (g/(mg·h)) is the adsorption rate constant of the pseudo-second-order.

#### 2.3.3. Adsorption Isotherms

Langmuir and Freundlich isotherms, which have been frequently used to model the adsorption of pollutants on adsorbents, were applied to describe the adsorption equilibrium data of BPA on MFMs [47,48].

The Langmuir isotherm equation is shown as follows in Equation (4).

$$\frac{1}{q_e} = \frac{1}{K_L q_{max}} \cdot \frac{1}{C_e} + \frac{1}{q_{max}} \tag{4}$$

The calculation of Freundlich model was performed by the following Equation (5):

$$\ln q_e = \ln K_F + \frac{1}{n} \ln C_e \tag{5}$$

where $q_e$ (mg/g) and $q_{max}$ (mg/g) are the equilibrium adsorption capacity and the maximum sorption capacity of adsorbent, respectively; $C_e$ (mg/L) is the equilibrium concentration of adsorbent; $K_L$ (L/mg) is the Langmuir constant; $K_F$ ((mg/g)/(mg/L)$^n$) is the Freundlich constant; n is the indicate heterogeneity factor.

### 2.3.4. Adsorption Site Energy Distribution Theory Analysis

The site energy distribution theory (SEDT) is a theoretical method to study the adsorption mechanism from the perspective of energy, which can provide information about the energies of the adsorption system, such as low-, average-, and high-energy adsorption sites [49]. Therefore, the analysis of site energy distribution was helpful to better understand the adsorption mechanism of BPA in water by MFMs and provides a more comprehensive theoretical basis. The energy distribution of adsorption sites for heterogeneous adsorbents can be calculated by Equation (6):

$$q_e(C_e) = \int_0^{+\infty} q_h(E, C_e) F(E) dE \tag{6}$$

where $q_h(E, Ce)$ and $F(E)$ are the homogeneous isotherm and site energy frequency distribution over local adsorption sites with adsorption energy E, respectively.

The condensation approximation proposed by Cerofolini has been widely used in practice as follows in Equation (7) [50].

$$C_e = C_s \exp\left(-\frac{E - E_s}{RT}\right) = C_s \exp\left(-\frac{E^*}{RT}\right) \tag{7}$$

where $C_s$ (mg/L) is the maximum solubility of the solute, and $E_s$ (kJ/mol) is the adsorption energy at concentration $C_s$. R, T, and $E^*$ are the universal gas constant, the absolute temperature (K), and the adsorption energy difference between solute and solvent on the adsorbent surface based on reference point $E_s$, respectively.

By correlating the energy distribution of the adsorption isotherm sites, an approximate SED function $F(E^*)$ could be obtained, as shown in Equation (8):

$$F(E^*) = \frac{-dq(E^*)}{dE^*} \tag{8}$$

where $F(E^*)$ and $q(E^*)$ (mg/L) are the energy distribution function of the adsorption site and the solute adsorption concentration, respectively.

According to Langmuir model Equation (4) and formulas (7) and (8), the distribution of adsorption sites can be obtained as follows:

$$F(E^*) = \frac{q_m K_L C_s}{RT} \exp\left(-\frac{E^*}{RT}\right) \left[1 + K_L C_s \exp\left(-\frac{E^*}{RT}\right)\right] \tag{9}$$

## 3. Results and Discussion

A preliminary study was carried out to investigate the effect of different of MFMs on the adsorption of BPA. As shown in Figure S1, the adsorption efficiencies of different MFMs were dramatically different, and the PTFE MFMs exhibited an extremely low adsorption efficiency. The main reason is that the PTFE MFMs are highly hydrophobic, leading to the BPA aqueous solution being hard to approach their surface. Thus, in the following experiment, PTFE MFMs were excluded for further study.

### 3.1. Adsorption Kinetic

The adsorption kinetic data of BPA on three MFMs are shown in Figure 2. As depicted in Figure 2, the uptakes of BPA on membranes increased dramatically in the first 2 h, and then, the adsorption rate slowed down with time. The adsorption equilibrium was reached within 5 h for all evaluated membranes. The rate-limiting step is a physical process for the pseudo-first-order model, but it depends on chemical processes for the pseudo-second-order model, which involves the adsorption force between the adsorbent and adsorbate [51]. Therefore, to study the rate-limiting steps and the mechanism of BPA adsorption on membranes, the pseudo-first-order model and the pseudo-second-order model were applied to fit the kinetic data (Figure 2). The parameters of these models are presented in Table 2. According to the graphical interpretation (Figure 2) together with judging the value of $R^2$, the adsorption data of BPA for the membranes fitted well with the pseudo-first-order model. The parameter $K_1$ in the pseudo-first-order model is frequently used to describe how fast the adsorption equilibrium is achieved, and the larger adsorption rate constant $K_1$ usually represents quicker adsorption [52]. The result dictated that the adsorption rate of the PVDF MFMs was the fastest, followed by PA and NC MFMs.

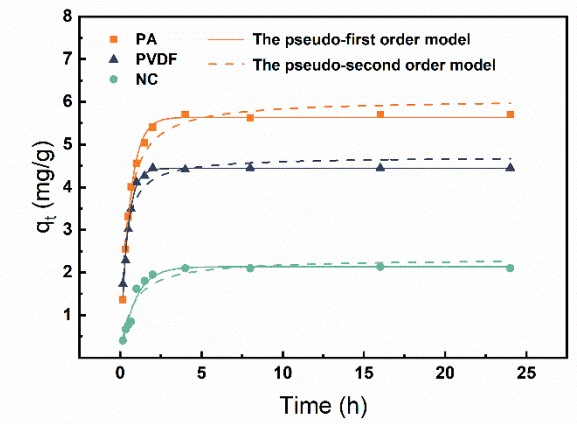

**Figure 2.** The kinetic adsorption model fitting of the pseudo-first-order model and pseudo-second-order model for the adsorption of BPA onto three MFMs.

**Table 2.** The kinetic adsorption constants of BPA by three MFMs.

| Membrane Material | Pseudo-First-Order Model | | | Pseudo-Second-Order Model | | |
|---|---|---|---|---|---|---|
| | $K_1$ (1/h) | $q_e$ (mg/g) | $R^2$ | $K_2$ g/(mg·h) | $q_e$ (mg/g) | $R^2$ |
| PA | 1.739 | 5.637 | 0.996 | 0.625 | 6.063 | 0.965 |
| PVDF | 2.373 | 4.442 | 0.986 | 0.810 | 4.717 | 0.938 |
| NC | 1.086 | 2.134 | 0.967 | 0.419 | 2.320 | 0.916 |

### 3.2. Adsorption Isothermal

Langmuir and Freundlich models were applied to fit the isotherm data to evaluate the maximum adsorption capacity, homogeneity, and heterogeneity in the surface of three MFMs and the adsorption mechanism. Figure 3 illustrates the modeling results of BPA adsorption on three MFMs based on Langmuir and Freundlich isotherms, together with relevant parameters summarized in Table 3. From the results, the Langmuir model agreed better than the Freundlich models judging from the "$R^2$" values ($R^2 > 0.99$). The Langmuir isotherm is generally performed to assess monolayer chemical-mediated adsorption on a homogeneous surface [53]. The results showed that the adsorption of BPA by the three MFMs was more inclined to monolayer adsorption. In addition, the $R^2$ of the Freundlich model was also excellent, which may account for the uneven distribution of energy on the membrane surface in the experiments. The Freundlich model is mainly used to study multimolecular adsorption, which can better reflect the heterogeneity of the adsorbent

surface [54]. Combining the correlations of the two models showed that the inhomogeneous distribution of energy on the membrane surface and the adsorption of BPA on the three MFMs were more biased towards monolayer adsorption in the non-homogeneous system.

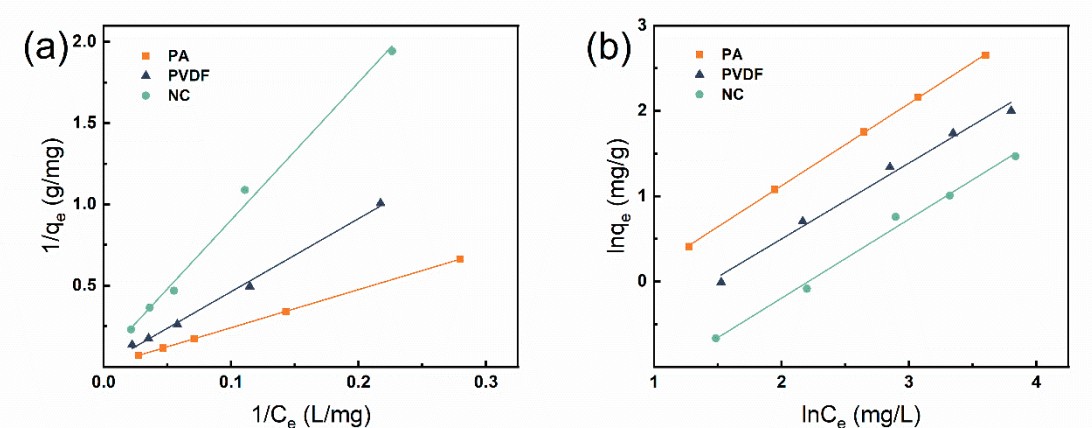

**Figure 3.** The isothermal adsorption model fitting of Langmuir (**a**) and Freundlich (**b**) for the adsorption of BPA onto three MFMs.

**Table 3.** The isothermal adsorption constants of BPA by three MFMs.

| Membrane Material | Langmuir Model | | | Freundlich Model | | |
|---|---|---|---|---|---|---|
| | $K_L$ (L/mg) | $q_{max}$ (mg/g) | $R^2$ | $K_F$ (mg/g)/ (mg/L)$^n$ | n | $R^2$ |
| PA | 0.0026 | 161.29 | 1.000 | 0.448 | 1.038 | 0.999 |
| PVDF | 0.0027 | 80.000 | 0.996 | 0.278 | 1.125 | 0.989 |
| NC | 0.0065 | 18.018 | 0.994 | 0.129 | 1.080 | 0.993 |

Based on the data of the Langmuir model, it was clear that PA MFMs had an apparently high BPA adsorption with the maximum adsorption capacity ($q_m$) up to 161.29 mg/g, while the $q_m$ of NC and PVDF MFMs were 18.02 and 80.00 mg/g, respectively. It showed that PA MFMs exhibited an evident adsorption capability of BPA that far exceeded other kinds of MFMs in the experiments. In the data of the Freundlich model, $K_F$ indicates the adsorption capacity, and PA MFMs > PVDF MFMs > NC MFMs, which is consistent with the results of the Langmuir model. In Table 4, some membrane materials are listed for the comparison of BPA-removal effectiveness. It is worth noting that the removal effect of PA MFMs is commendable.

**Table 4.** Comparison of maximal adsorption capacity of BPA by various membranes.

| Membrane Material | Membrane Type | $q_m$ (mg/g) | Ref |
|---|---|---|---|
| PA | MF | 161.29 | This study |
| PVDF | MF | 80.00 | This study |
| NC | MF | 18.02 | This study |
| PTFE | MF | 1.56 | This study |
| NNM | NF | 91.30 | [55] |
| CA-P-CDP | NF | 50.37 | [56] |
| PP-g-SA-HEA-PVDF | Composite membrane | 26.67 | [57] |
| β-CD/CS/PVA | NF | 352.17 | [58] |
| CDGO | Composite membrane | 25.50 | [59] |
| SRt-PAN | NF | 17.50 | [60] |

*3.3. Site Energy Distribution Analysis*

In order to deepen the understanding of the adsorption potential energy of solute BPA removal by three MFMs, the adsorption of BPA by MFMs was further analyzed using the theory of adsorption potential energy distribution. The site energy $E^*$ of BPA adsorption on three types of MFMs as a function of $q_e$ are shown in Figure 4. The $R^2$ data (Table 3) using the Langmuir model were the most desirable for the experimental analysis of adsorption isotherms, so the site energy analysis was performed based on the Langmuir model, and the relevant data obtained are shown in Table 5.

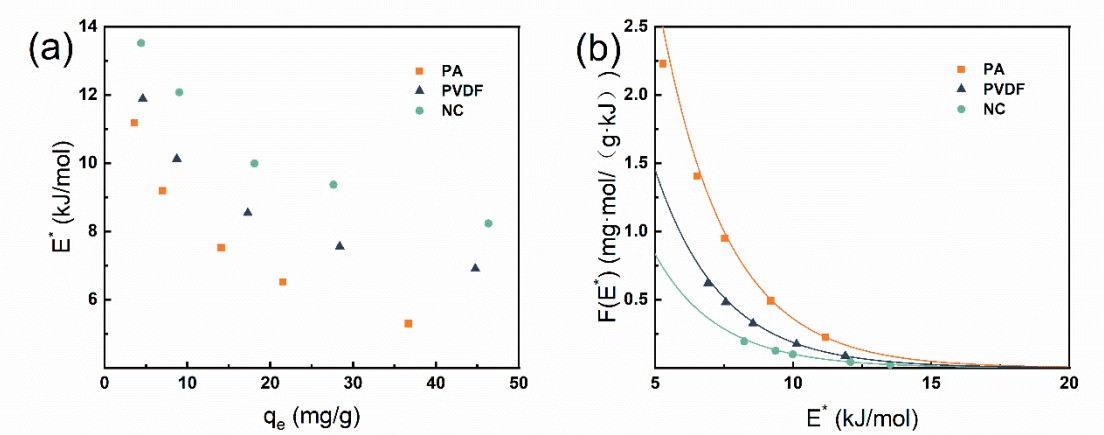

**Figure 4.** Site energy distribution of BPA adsorption by different MFMs: Adsorption energy vs. adsorption amount (**a**) and frequency distribution vs. adsorption energy (**b**).

Figure 4a shows that the adsorption energy (E*) decreased rapidly with increasing $q_e$, which verified the possible inhomogeneous properties of SED and the existence of a limited number of high-energy adsorption sites on the MFMs [61]. As the BPA concentration increased, the BPA molecules first attached to the high-energy sites until the high-energy sites were completely occupied, and then, they shift to the low-energy sites for adsorption [62]. The high-energy and low-energy regions often correspond to the low- and high-concentration regions, respectively. As shown in Figure 4b, the energy distribution of the adsorption sites fitted by the three MFMs for the BPA adsorption process belonged to the L-class shape, which indicated that the adsorbent had a high affinity for the adsorbate in the low concentration range [63]. By comparing the energy range of the three MFMs in Table 5, the adsorption process of BPA by PA MFMs was more inclined to the low potential energy region, which means that PA MFMs had a better adsorption affinity for the adsorption of BPA. By definition, the area formed between the bottom of the curve and the coordinate axis in Figure 4b shows the number of available adsorption sites in a specific energy range [64]. Although the shapes of the distribution functions for BPA adsorption were similar for the three MFMs, there were significant differences in the areas under their curves. Most of the adsorption sites were obtained by BPA on PA MFMs, followed by PVDF and NC MFMs, which was consistent with the previous analysis that the adsorption capacity of BPA was PA > PVDF > NC.

**Table 5.** The parameters of the adsorption energy distribution of BPA on different MFMs.

| Membrane Material | The Energy Range | The Average Site Energy | The Site Energy Heterogeneity |
|---|---|---|---|
| | E (kJ/mol) | μ (E*) (kJ/mol) | $\sigma_e^*$ (kJ/mol) |
| PA | 5.298–11.173 | 7.940 | 2.059 |
| PVDF | 6.609–11.886 | 9.004 | 1.802 |
| NC | 8.230–13.514 | 10.635 | 1.907 |

### 3.4. Rapid Filtration Adsorption and Equilibrium Adsorption

To provide the level of adsorption required by water treatment, the adsorption property of the three types of MFMs by rapid filtration process and equilibrium adsorption process was evaluated and compared. As can be seen in Figure 5, the BPA removal efficiency of PA MFMs in filtration was obviously better than that of the other two MFMs, while the NC membrane was the worst, which is consistent with the previous thermodynamic and kinetic experimental results. Compared to equilibrium adsorption, the rapid filtration adsorption showed that the removal efficiency of BPA molecules was higher, with the most obvious change for PA MFMs. There may be several reasons: Firstly, during filtration, the solution had a natural pressure on the membrane, and the pressure drove the rapid diffusion of BPA molecules to the membrane surface, so the diffusion of BPA solutes from the solution to the membrane surface was faster during filter adsorption. Furthermore, the filtration process drove the BPA molecules in the solution to bind to the membrane more fully, occupying more adsorption sites on the membrane surface and inside the membrane. Therefore, the removal efficiency was greatly enhanced. In addition, the contact time between the BPA solution and the membrane during the filtration process was shorter, and the filtration may have been completed before desorption occurred. However, for equilibrium adsorption, the adsorbed solute may be desorbed and finally reach the equilibrium of adsorption and desorption. Therefore, the removal efficiency of BPA was higher at rapid filtration than at the equilibrium adsorption, which would facilitate the removal of BPA in an aqueous solution by filtration adsorption.

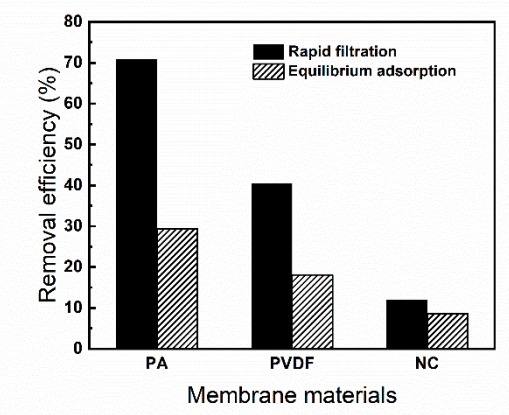

**Figure 5.** The removal efficiency of BPA on different MFMs by rapid filtration and equilibrium adsorption. (The removal efficiency of different MFMs immersed in 50 mL of 10 mg/L BPA solution for 24 h was compared with the removal efficiency of different MFMs filtering 50 mL of 10 mg/L BPA solution.).

### 3.5. Adsorption Mechanism

According to the adsorption isotherm and kinetic analysis, three MFMs exhibited a significantly different adsorption capacity and adsorption rate for BPA. The order of adsorption capacity was PA MFMs ($q_{max}$ = 161.29 mg/g) > PVDF MFMs ($q_{max}$ = 80.00 mg/g) > NC MFMs ($q_{max}$ = 18.02 mg/g). The order of adsorption rate was the PVDF MFMs ($K_1$ = 2.373 h$^{-1}$) > PA MFMs ($K_1$ = 1.739 h$^{-1}$) > NC MFMs ($K_1$ = 1.086 h$^{-1}$). In the batch adsorption process, both the chemical composition and physical structure of absorbents can affect solute adsorption. The physical structure has an effect on the physical adsorption resulting from a specific surface area [65]. The adsorption mechanism, such as electrostatic interaction, hydrogen bonding, and hydrophobic partitioning, can be attributed to the different chemical composition of the absorbent towards the target solute [66]. In this study, the target solute BPA was a hydrophobic compound (log $K_{ow}$ = 3.32) [57]. Hydrophobic partitioning is expected to occur in aqueous solutions in the presence of the hydrophobic microdomain of absorbents. The phenolic hydroxyl and alkylene moieties on the BPA

molecule can act as proton donors and bind to the electronegative oxygen and nitrogen atoms via intermolecular hydrogen bonds. BPA molecules undergo a minimal dissociation in pH-neutral conditions due to their high acid dissociation constant (pKa = 9.6–10.2), therefore, electrostatic interactions would be limited [67].

As shown in Figure 6, the SEM of PA and PVDF MFMs had a similar surface morphology, but there was an obvious adsorption difference between PA and PVDF MFMs. Furthermore, NC MFMs with a fibrous-like pore wall seem to have a higher specific surface area; however, the adsorption capacity and rate by NC MFMs was the lowest. These indicated that the physical structure was not the main factor affecting the BPA adsorption. The different adsorption capacities of the three types of MFMs can be attributed to their different chemical composition.

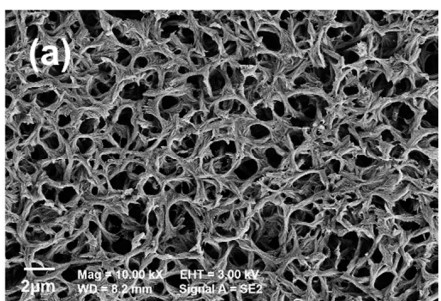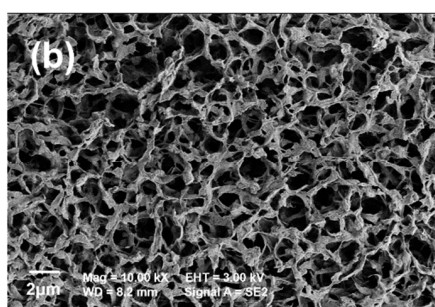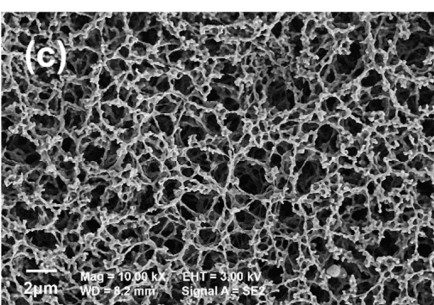

**Figure 6.** SEM image of the surface morphology of the PA MFMs (**a**), PVDF MFMs (**b**), NC MFMs (**c**).

The chemical composition and content of three MFMs were further analyzed by EDS and elemental mapping in Figure 7. The element composition of PA MFMs and NC MFMs is consistent with their molecular structure formula in Table 1, and their element percentage is basically consistent with the content of elements in the structure, respectively. From the chemical composition of three MFMs, PA and NC MFMs have the same element composition, but the O and N content of NC is higher than PA. About 3.4% (atomic percentage) of O content is on the PVDF MFMs, which is because the PVDF MFMs were modified for improving hydrophilicity. Because O or N elements on the MFMs possess a lone electron pair, they can be proton acceptors and bind to proton donors. There are two phenolic hydroxyls in BPA molecules, which are known to be active proton donors [34]. Therefore, hydrogen bonding may occur between the BPA and the MFMs.

Actually, the carbonyl oxygen atom presenting in the amide group of PA has a high negative atomic charge and can combine with the amine proton on the adjacent amide group to form inter-molecular hydrogen bonds. When interacting with a foreign molecule with a strong proton donor, the existing hydrogen bonds in PA change to form preferential hydrogen bonds with the molecule in contact [68]. After adsorption of BPA, there were obvious fluctuations in the peak at 1635 cm$^{-1}$ and 1539 cm$^{-1}$ in Figure 8a. The change in band intensity after the adsorption of BPA may be due to the hydroxyl group (-OH) of the BPA molecule as a hydrogen bond donor and the amide group of PA as a hydrogen bond acceptor through intermolecular interactions carried out by hydrogen bonding [69]. The FTIR spectra of PVDF MFMs proved that hydroxyl groups (~3500 cm$^{-1}$) and carboxyl groups (~1700 cm$^{-1}$) were modified on PVDF MFMs in Figure 8b. Compared with the membranes before the adsorption of BPA, it was obvious that a new ester group characteristic peak appeared at 1713 cm$^{-1}$, which may be a combination of surface (C=O) vibrational stretching and the hydroxyl group of the BPA molecule to form intermolecular hydrogen bonds [70]. Nonetheless, the characteristic adsorption peaks of the NC MFMs before and after adsorption can be seen in the FTIR spectra of Figure 8c with almost no changes. NC is a polymer produced commercially by reacting purified cellulose from plants with nitric acid and replacing the cellulose hydroxyl group with nitrate [71]. The nitric acid and hydroxyl groups in NC made it easy to interact with solute molecules by electrostatic interaction and hydrogen bonds. Nevertheless, electrostatic interactions would be limited, as BPA

molecules only undertake microscopic dissociation in pH-neutral conditions, as shown by its high acid dissociation constant (pKa = 9.6–10.2) [67]. The peaks responding to stretching vibration (~2910 cm$^{-1}$) and deformation vibration (~1371 cm$^{-1}$) of the C-H bond of NC had no obvious change. It proved that there are no obvious interactions between NC and BPA molecule [72]. Hydrogen bonding forces may occur between BPA and NC MFMs, but it is difficult to detect because of the low adsorption capacity of NC MFMs for BPA.

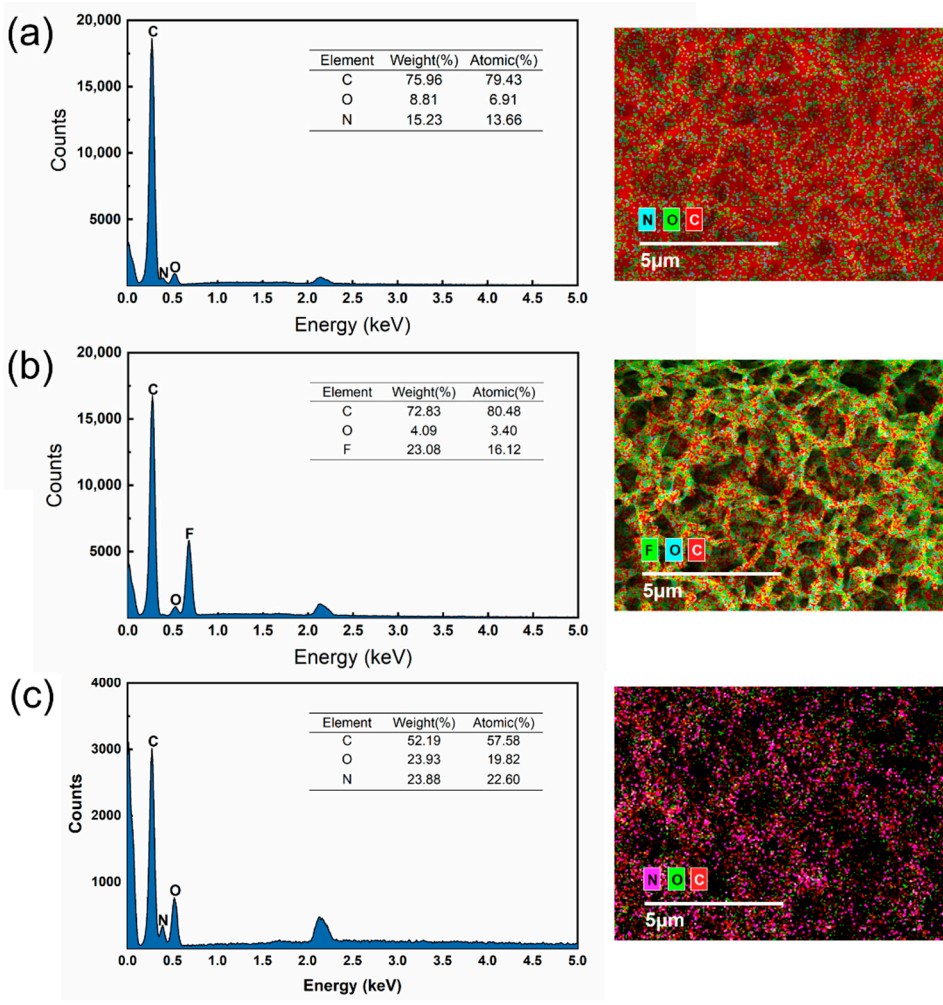

**Figure 7.** EDS and elemental mapping of the PA MFMs (**a**), PVDF MFMs (**b**), and NC MFMs (**c**).

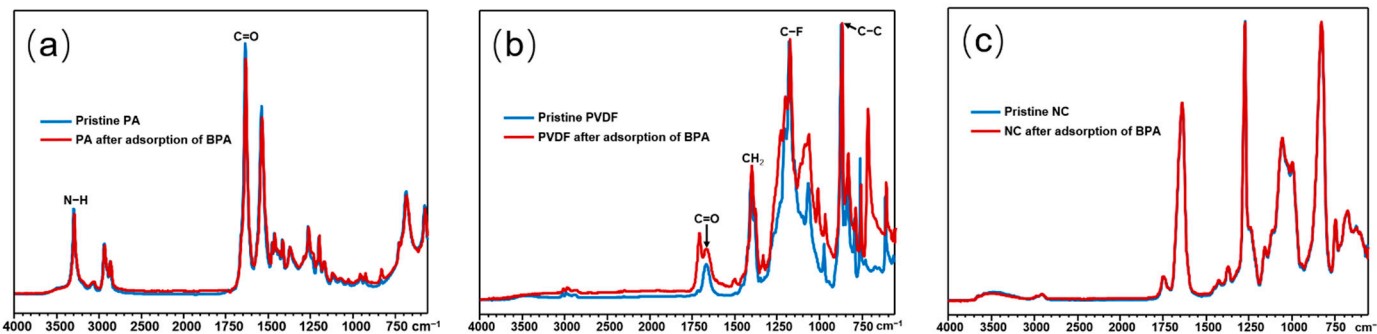

**Figure 8.** FTIR spectrum of the PA MFMs (**a**), PVDF MFMs (**b**), and NC MFMs (**c**).

The FTIR results proved that hydrogen bonding was generated when BPA contacted the PA and PVDF MFMs. In theory, the number of proton acceptors was NC MFMs > PA MFMs > PVDF MFMs. However, the adsorption of BPA did not agree with the number

of proton acceptors. Although hydrogen bonding played an important role for BPA on membranes, there should be other interactions between BPA and membranes.

Hydrophobic partitioning is expected to occur in aqueous solutions in the presence of the hydrophobic microdomain of adsorbents due to the hydrophobic nature of BPA. The hydrophobicity or hydrophilicity of absorbents can be evaluated by the static contact angle and water absorption coefficient. The static contact angle reflects the instantaneous interaction between the absorbent's surface and water, while the water absorption coefficient reflects the hydrophilic properties of the absorbent's surface and interior. Three MFMs have the similar contact angle of less than 90 degrees in Table S1, which means that all MFMs' surfaces are hydrophilic [73]. However, there was an obvious water adsorption difference among the three MFMs. As shown in Figure 9, NC MFMs had the highest water absorption coefficient, with an average water absorption rate as high as 213.81%. The average water absorption coefficient of the PA MFMs was 84.41%, and of PVDF MFMs, it was 80.99%. These indicated that the NC MFMs had the highest hydrophilicity, leading to the adsorption sites being hindered by water molecules. This can explain why NC MFMs had the lowest adsorption for BPA than PA and PVDF MFMs. Meanwhile, these suggested that PA and PVDF MFMs are more hydrophobic than NC MFMs, which was further proved by the EDS mapping in Figure 7, showing that the hydrophobic microdomain existed in MFMs.

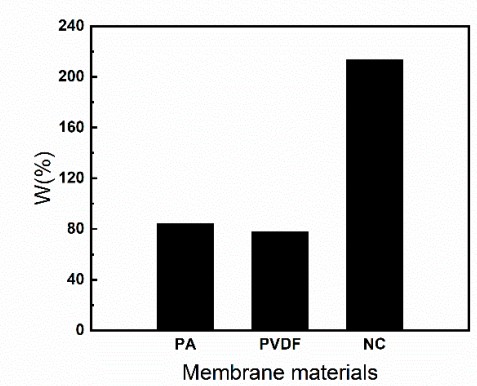

**Figure 9.** The coefficient of water absorption (W) of three MFMs.

The PA MFMs used were aliphatic PA. The main chain of aliphatic PA molecules included a large number of amide groups (-CO-NH-) as hydrophilic moieties and separated by methylene (-$CH_2$-) sequences of hydrophobic moieties. As shown in Figure 7a, green and blue dots represent oxygen and nitrogen atoms, respectively, as the hydrophilic microdomain, which are slightly separated from the carbon chain represented by red dots as the hydrophobic microdomain in PA MFMs. Nitrocellulose is a polymer with a lot of nits and hydroxyl groups [71]. PVDF MFMs have a low surface energy and hydrophobicity [32]. F atoms represented by green dots are wrapped around C atoms as the hydrophobic microdomain, and few O atoms are represented by blue dots, as hydrophilic microdomains are dispersed in the PVDF MFMs. As shown in Figure 7, the hydrophobic microdomain on PVDF seems to be larger than PA and PA larger than NC, which was consistent with the water absorption coefficient of the three MFMs. Therefore, the hydrophobic interaction between BPA and PVDF MFMs may be strongest, followed by PA and NC MFMs. However, the higher hydrophobicity of PVDF MFMs did not exhibit a higher adsorption capacity than PA MFMs, which implied that hydrophobic interaction is not the predominant driving mechanism for the BPA adsorption on MFMs. Combined with the results of FTIR, hydrogen bonding was generated when BPA contacted the PA and PVDF MFMs. The proton acceptors (O, N content) of PA MFMs are far more than PVDF MFMs based on the elemental analysis and FTIR. Thus, it is possible that the adsorption driven by hydrogen bonding of PA is stronger than PVDF. Indeed, hydrogen bonding energy is usually in the range of 8–50 kJ/mol, which is much higher than hydrophobic interaction [74,75]. The high-energy

hydrophilic microdomain hydrogen site was occupied preferentially. From the point of energy, the hydrogen bond offered more energy to overcome the energy barrier of the adsorption and diffusion process. These agreed with the results of the site energy distribution analysis mentioned above. Thus, it makes sense that the adsorption capacity of PA MFMs is far more than PVDF MFMs. Nevertheless, the adsorption rate of PVDF MFMs is faster than PA and NC MFMs. The adsorption rate is related to the mass transport of the solute across MFMs, which may involve three major consecutive processes [76]: (1) diffusion from the water phase into the pore of the membrane; (2) sorption onto and then diffusion across the membrane; (3) desorption from the permeate side of the membrane. As the water molecules occupied the adsorption sites on hydrophilic NC MFMs, the BPA molecules struggled to access the membrane. The adsorption of BPA on NC MFMs was limited. Due to the hydrophobic microdomain on PA and PVDF MFMs and the hydrophobic nature of BPA, the hydrophobic partitioning may contribute to BPA diffusing to the surface of the MFMs. Considering the diversity of hydrophobicity of MFMs, these may be able to explain why the adsorption rate of PVDF is faster than PA and NC. Owing to the weak hydrophobic interaction, desorption can easily occur after adsorption. It was proven from the results of rapid filtration adsorption and equilibrium adsorption that the removal efficiency of BPA by filtration adsorption was larger than the equilibrium adsorption, because adsorption occurs rapidly, but desorption has not yet started during the rapid filtration.

Based on the analysis above, hydrogen bonding and hydrophobic partitioning are important factors in determining the adsorption capacity and adsorption rate of MFMs. We speculate that hydrogen bonding contributes to the BPA molecules' strong binding, but hydrophobic interaction is conducive to the BPA molecules approaching the surface of the membrane. The interaction mechanism between membranes and BPA is shown in Figure 10. The highly hydrophilic surface of NC MFMs makes most of the adsorption sites occupied by water molecules, and the highly hydrophobic surface of PTEF MFMs makes BPA molecules surrounded by the water molecules, struggle access the membrane surface, which leads to these membranes having a low adsorption capacity. However, the hydrophilic microdomain and hydrophobic microdomain have a micro-separation for PA and PVDF MFMs, which result in the adsorption capacity of PA and PVDF MFMs far more than NC and PTFE MFMs. The hydrophilic microdomain providing hydrogen bonding sites and the hydrophobic microdomain providing hydrophobic interaction with the hydrophobic solute, work together to promote the adsorption of BPA on the membrane surface. Due to the hydrogen bonding force being greater than the hydrophobic force, more hydrogen bonding sites on the hydrophobic surface result in a higher adsorption capacity, but hydrophobic interaction contributes to improve the adsorption rate. Hence, the distribution of the hydrophilic microdomain and hydrophobic microdomain on the surface of the membrane can influence the adsorption capacity and adsorption rate for BPA or its analogues.

Since hydrogen bonding and hydrophobic partitioning are important factors in determining the adsorption capacity and adsorption rate of MFMs, this means that it may be sensitive to certain water chemistry parameters [77]. In fact, in actual wastewater, there may be water chemistry parameters such as salinity, pH, and natural organic matter (NOM) that have an effect on the adsorption process of MFMs, such as the partitioning of hydrophobic solutes in water, which may be affected by salinity, and NOM may have competitive adsorption with BPA [56,77]. However, the effect of water chemistry parameters on the adsorption of BPA by MFMs was not explored in depth in this work and will be discussed further in our subsequent studies.

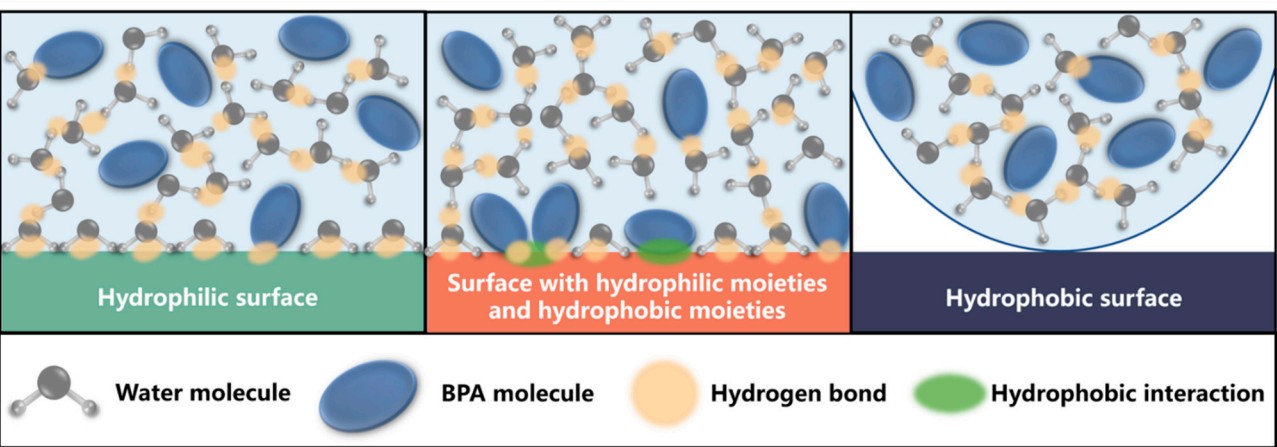

**Figure 10.** The interaction mechanism between BPA and membranes.

## 4. Conclusions

Four specific MFMs with different chemical compositions were applied to analyze their adsorption behavior and adsorption mechanism to evaluate the feasibility of MFMs as adsorbents to remove BPA. There are major conclusions from this study: (1) There is obvious different adsorption behavior among these MFMs. The maximum adsorption capacity of PA MFMs reached 161.29 mg/g, but PVDF MFMs exhibited the maximum adsorption rate. A further site energy distribution analysis suggested that PA MFMs had the greatest adsorption sites, followed by PVDF and NC MFMs. (2) The removal efficiency of BPA was higher at rapid filtration than at the equilibrium adsorption, which would facilitate the removal of BPA in an aqueous solution by filtration adsorption. (3) The hydrophilic microdomain provided hydrogen bonding sites and the hydrophobic microdomain provided hydrophobic interaction, which were important driving mechanism for the BPA adsorption on membranes. (4) The adsorption of BPA by either highly hydrophilic or highly hydrophobic MFMs is unfavorable. The MFMs with a micro-separation of the hydrophilic microdomain and hydrophobic microdomain can promote the adsorption of BPA. The adsorption capacity and adsorption rate can be adjusted by adjusting the ratio of the hydrophilic microdomain to hydrophobic microdomain on the surface of membranes.

**Supplementary Materials:** The following supporting information can be downloaded at https://www.mdpi.com/article/10.3390/w14152306/s1. Figure S1: The schematic diagram of the experimental apparatus of rapid filtration adsorption; Figure S2: Comparison of the maximum adsorption capacity calculated by the Langmuir model for isothermal adsorption of four MFMs; Table S1: The static contact angle of three MFM.

**Author Contributions:** J.S.: Conceptualization, Methodology, Supervision. X.J.: Investigation, Data curation, Writing—Original draft. Y.Z.: Investigation, Methodology. J.F.: Supervision, Writing—Reviewing and Editing. G.Z.: Writing—Reviewing and Editing. All authors have read and agreed to the published version of the manuscript.

**Funding:** This research was funded by the Natural Science Foundation of Chongqing, China (Project No. cstc2020jcyj-msxm X0928) and the Natural Science Foundation of China (Project No. 41977337).

**Data Availability Statement:** Not applicable.

**Conflicts of Interest:** The authors declare that they have no known competing financial interest or personal relationship that could have appeared to influence the work reported in this paper.

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
