# Peer review of "Microfiltration Membranes for the Removal of Bisphenol A from Aqueous Solution: Adsorption Behavior and Mechanism"

_water, doi:10.3390/w14152306_

Round 1
Reviewer 1 Report
This study deals with the use of microfiltration membranes for the removal of bisphenol A from aqueous solutions. The study is interesting; however, the authors are advised to attend the following observations.
- It is suggested in the summary to report the values in mg g-1 and % of maximum adsorption capacity and adsorption efficiency or adsorption rate respectively, in the sentence between lines 12 - 14.
- There are acronyms that are defined more than once in the manuscript, e.g., PA, PVDF, NC, PTFE, this should be reviewed.
- Mention table 1 in section 2.1 to justify the placement of this table in that section. In addition, it is suggested to add citations to support the assertions made in the "Performance" column.
- In table 3, it is recommended to include a column with the maximum experimental BPA retention capacity and to compare the result between them. Furthermore, it is suggested to compare the best results obtained in this study with similar systems already reported in the scientific literature, in order to highlight the novelty.
- It is suggested to add the chemical structure of BPA in the manuscript.
- In Figure 2 it is suggested to check the x and y axes (they are changed), the axes do not agree with the isothermal models presented.
- Figure 7c should report the main link vibrations in the corresponding signals, as shown in figure 7a and 7b.
- Figure 9 should be quoted in the text. In addition, Figure 9 should define what the white colour surrounding the BPA molecules represented by a blue ellipse represents.
- It is recommended to standardise the format of all the figures, specifically the typeface and font size used in each figure so that they are legible for the reader, and to eliminate the grey background in the corresponding figures.
- On page 9 line 314 it is recommended to complement the above by measuring with software the average pore size of the micrographs and report.
- There is a lack of citations to support the assertions in lines 37, 48, 55, 63 and 358.
- Add the pKa of the functional groups involved in BPA retention to line 358.
- It is suggested that sections 3.1 to 3.4 be enriched with more references to support each assertion made. However, in general, it is suggested to support the assertions made in the results and discussion section with more literature.
- It is suggested to reorder the whole section 3. results and discussion, starting with the characterisation of their materials, continuing with the application in BPA retention and ending with the mechanism of BPA retention.
- It is suggested to complement their analyses by incorporating one that demonstrates the effect of pH on BPA retention.
Finally, authors are encouraged to carefully address the above comments and corrections to improve the quality of this manuscript
Reviewer 2 Report
Development of efficient water and wastewater treatment methods has high importance for the practice. Among the water pollutants, BPA has high environmental and health risk; therefore, processes for removal of BPA have high relevance. Membrane separation method can answer for these challenges. Investigation of BPA adsorption on microfiltration membranes made from different materials can provide interesting data and information for the readers. The manuscript is generally well structured. Introduction summarizes well the relevance of the study. Applied methods are adequate. Tables and figures represent well the experimental results. Results are discussed in details with relevant references. The manuscript has high scientific quality, in general.
Comments
Please provide detailed characteristics of membranes (pore size, pH and temperature tolerance etc).
It is not clear why use 20 mg/L BPA concentration for adsorption kinetic tests. Please explain it.
The visibility of Figure 1, 2 and 8 is too poor (mainly axis titles, labels). Please improve it.
Please discuss briefly the matrix effect on BPA adsorption (other components in real samples/effluents)
Reviewer 3 Report
In this work, four specific MFMs with different chemical compositions were applied to analyze their adsorption behavior and adsorption mechanism to evaluate the feasibility of MFMs as adsorbents to remove BPA. I think this work is praiseworthy for its rigorous experimental data and wealthy explanations. Detailed comments are listed as follows.
Recommendation: Publish after major revisions noted.
1. The main content in the abstract should include important conclusions and shining results of this study, including the representative data, the underlying mechanisms and the future enlightenment. The authors should rewrite this part.
2. The experimental process should have been accompanied with Zeta potential studies at different pH, identifying the PZC (point of zero charge).
3. Part of the introduction should be improved. I recommend that authors add some new references concerning novel materials in the introduction.
10.1021/acsami.1c22035, 10.1016/j.clay.2018.12.017
4. Authors should provide the economic study of the production for commercial viability.
5. The author should list a table to compare the adsorption effect of materials with other literatures.
6. Other Problems: there are a few mistakes in the format of references, which need further improvement.
Round 2
Reviewer 3 Report
The manuscript was improved in its entirety, for which it is confirmed that the authors considered all the observations, comments and suggestions made by the reviewers. I believe that the article can be published.